# Dietary Curcumin Supplementation Enhanced Ammonia Nitrogen Stress Tolerance in Greater Amberjack (*Seriola dumerili*): Growth, Serum Biochemistry and Expression of Stress-Related Genes

**Jiawei Hong [1,2,3], Zhengyi Fu [1,2], Jing Hu [1,2], Shengjie Zhou [1,2], Gang Yu [1,2] and Zhenhua Ma [1,2,*]**

[1] Key Laboratory of Efficient Utilization and Processing of Marine Fishery Resources of Hainan Province, Sanya Tropical Fisheries Research Institute, Sanya 572018, China

[2] Tropical Aquaculture Research and Development Center, South China Sea Fisheries Research Institute, Chinese Academy of Fishery Science, Sanya 572018, China

[3] State Key Laboratory of Marine Environmental Science, College of Ocean and Earth Sciences, Xiamen University, Xiamen 361005, China

[*] Correspondence: zhenhua.ma@hotmail.com

**Abstract:** This study was conducted to determine whether curcumin has a positive effect in greater amberjack (*Seriola dumerili*), especially the ammonia nitrogen stress tolerance ability. The results showed that the stress recovery process of digestive enzymes amylase and trypsin, as well as absorptive enzymes $Na^+/K^+$-ATPase, $\gamma$-GT and CK, was accelerated. Lysozyme activity increased in the fish fortified with both curcumin diets. Aspartate aminotransferase activity restriction was activated at a low curcumin level. However, alanine aminotransferase activity restriction happened only at 0.02% dietary curcumin. Facilitation of lipid metabolism by curcumin was very clear, as triglyceride and total cholesterol content was basically maintained at the original level or even showed a slight decrease after recovery. HSP70 and HSP90 genes were not evidently stimulated to express in liver, kidney and spleen tissues. In addition, curcumin showed its inhibition capacity on IL1$\beta$ and IFN-$\gamma$ and a promoting effect on TGF-$\beta$1. The expression of NF-$\kappa$B1 decreased in a higher degree in fish fed with 0.02% dietary curcumin, while 0.01% dietary curcumin accelerated the recovery pace of C3 and lgT after stress. This study showed that dietary curcumin supplementation can enhance ammonia nitrogen stress tolerance in greater amberjack, and its application prospect can be confirmed.

**Keywords:** *Seriola dumerili*; curcumin; ammonia nitrogen tolerance

## 1. Introduction

Aquaculture has developed so far that intensification is thought to be a great solution to increase production and revenue. However, it comes at a cost, such as water pollution and health issues [1–5]. Furthermore, the appearance of antibiotic resistance caused by drug abuse has brought aquaculture into an adverse cycle [6]. Ultimately, once all these negative factors gradually accumulate to a certain extent, they can restrict the development of aquaculture production [7]. Greater amberjack (*Seriola dumerili*) is greatly favored in many countries and regions, and it is considered a significant species for promoting aquaculture diversification around the world [8,9]. Greater amberjack has a high commercial value due to its high quality of meat and large body size [10–13]. In addition, it also has excellent production characteristics, such as high feed utilization, a rapid growth rate and low mortality in any kind of culture method [8,14–16]. As a

consequence, its culture has been growing at a remarkable speed in recent years. However, as an important part of cultured fish, greater amberjack is also facing the same dilemma of the threat of water quality deterioration caused by high-density aquaculture as other species as a consequence of efforts to increase yield.

In addition to the efforts in the breeding of disease- and stress-resistant varieties as well as culture equipment and methods, the exploration of immune boosters in diet is also a key breakthrough in strengthening fish health; it is also another way to increase aquaculture production. In recent years, much research involving the great functions of vegetative dietary supplementation has emerged, especially into Chinese herbs [17]. Many essential effective ingredients of Chinese herbs have been screened and identified and have proved to be sufficient to become excellent promoters of immune functions in many species. Meanwhile, the sources of these herb extracts are natural and easily available [18–20]. These advantages further increase their research potential and value. Therefore, our study focused on vegetative immune boosters based on their previous performance reported in much research. For example, a dietary addition of a mixture of Astragalus and Lonicera extracts to *Nile tilapia* exhibited enhanced phagocytosis, as well as a rise in respiratory burst and lysozyme activity [21]. Similarly, black rockfish treated with green tea ethanol extract in their diet also witnessed an increase in lysozyme activity; moreover, stress recovery capacity was strengthened at the same time [22]. Ref. [23] found that an *Azadirachta indica* extract showed an effect equal to that of antibiotics on treating *Citrobacter freundii* infection in *Oreochromis mossambicus*. As to our study, the supplementation we tested is curcumin.

Curcumin is a polyphenol derived from the roots of *Curcuma longa* L. or other Curcuma spp. plants [24,25], which has a long history of use by humans [26]. It is well-known not only for the functions of colouring and flavouring [27], but also for its extensive health care effects, such as antioxidant actions, anti-inflammation, inhibition of carcinogenesis, oxygen radical scavenging, liver protection, etc. [28–32]. In view of its numerous positive effects, curcumin was soon studied and applied in mammals, and it has also attracted great attention in aquaculture in recent years. Curcumin and its active molecules have been proved to have effects on fish health similar to those on human and other mammals [33–35]. For example, Ji et al., found that 0.04% curcumin supplementation can reduce hepatic lipid deposition, improve antioxidant activity and increase PUFA of large yellow croaker; and finally, to eliminate the side effects caused by a high-fat diet [36]. Curcumin supplements in carp diet can increase the activity of antioxidant enzymes by promoting the release of Nrf2 and ultimately protect carp from liver damage caused by CCl4 [37]. Hemat et al., also found that dietary curcumin showed great performance in improving growth, feed utilization, oxidative status, immune responses, and disease resistance in *Nile tilapia* [38].

Based on all the known information, the application prospect of dietary curcumin supplementation in greater amberjack is highly anticipated. Thus, whether curcumin can show great performance in improving the health characteristic of greater amberjack is well worth investigating.

## 2. Materials and Methods

### 2.1. Experimental Fish and Feeding Experiment

The greater amberjack used in the study were bred by the Tropical Fisheries Research and Development Center, South China Sea Fisheries Research and Development Center, South China Sea Fisheries Research Institute, Chinese Academy of Fishery Science (Lingshui, Hainan, China). After being cultivated temporarily for one week, 135 individual fish appearing to be healthy and energetic and showing no injuries were selected and randomly divided into three experimental groups. Every treatment was in triplicates, and each triplicate had 15 fish. Initial mean weights of fish in each group were 149.02 ± 12.59 g, 153.51 ± 4.09 g and 151.78 ± 3.84 g, respectively. The experiment was

conducted in a circulating mariculture tank (5000 L). Water temperature, salinity, and the pH of the aquaculture seawater during the period were 27–31.5 °C, 35‰ and 7.5–8, respectively. Ammonia nitrogen content was maintained below 0.01 mg/L, nitrite content below 0.02 mg/L, and dissolved oxygen was always above 7.0 mg/L. The fish were fed to apparent satiety twice a day (at 8:00 am and 16:00 pm). Feces were siphoned out to avoid water quality deterioration at 1 h after feeding. The feeding experiment lasted eight weeks.

### 2.2. Experimental Diet

The addition amounts of curcumin in the three experimental diets were 0, 100 and 200 mg/kg, respectively. All diets were designed and produced by the Tropical Fisheries Research and Development Center, South China Sea Fisheries Research and Development Center, South China Sea Fisheries Research Institute, Chinese Academy of Fishery Science. The added curcumin (purity > 95%) was provided by Xi'an Feida biotechnology Co., Ltd. (Xi'an, China). The formulation of experimental diet is shown in Table 1.

**Table 1.** Formulation and proximate composition of the experimental diet.

| Ingredients | Proportion (g/kg) | | |
|---|---|---|---|
| | Control | 0.01% CUR [1] | 0.02% CUR [1] |
| Fish meal | 590 | 590 | 590 |
| Corn gluten meal | 70 | 70 | 70 |
| Soybean meal | 80 | 80 | 80 |
| Corn starch | 80 | 80 | 80 |
| Microcrystalline Cellulose | 50 | 50 | 50 |
| Fish oil | 70 | 70 | 70 |
| Phospholipids | 10 | 10 | 10 |
| [2] Vitamin premix | 5 | 5 | 5 |
| [3] Mineral premix | 5 | 5 | 5 |
| Choline chloride | 5 | 5 | 5 |
| Betaine | 5 | 5 | 5 |
| Carboxymethyl cellulose | 30 | 30 | 30 |
| Curcumin | 0 | 0.1 | 0.2 |
| Analyzed nutrients | | | |
| Dry Matter | 878.98 | 878.98 | 878.98 |
| Crude protein | 497 | 497 | 497 |
| Crude lipids | 126.98 | 126.98 | 126.98 |
| Crude ash | 106.85 | 106.85 | 106.85 |

[1] 0.01% CUR and 0.02% CUR indicate the treatment groups in which the proportion of curcumin supplementation in the diet are 0.01% and 0.02%. [2] Vitamin premix (mg/kg diet): vitamin A—9,000,000 (IU/kg diet), vitamin K3—600 (IU/kg diet), vitamin D—2,500,000 (IU/kg diet), vitamin E—500 (IU/kg diet), vitamin B—(B1 3200 (IU/kg diet), B2—10,900 (IU/kg diet), B5—20,000 (IU/kg diet), B6—5000 (IU/kg diet), B12—1160 (IU/kg diet), vitamin C—50,000 (IU/kg diet), phaseomannite—1500 mg/kg diet, calcium pantothenate—200 mg/kg diet, niacin—400 mg/kg diet, folic acid—50 mg/kg diet, biotin—2 mg/kg diet. [3] Mineral premix (mg/kg diet): KCl—70; KI—1.5; $MgSO \cdot 7H_2O$—300; $MnSO_4 \cdot 4H_2O$—3; $CuCl_2$—5; $ZnSO_4 \cdot 7H_2O$—14; $CoCl_2 \cdot 6H_2O$—0.5; $FeSO_4 \cdot 7H_2O$—15; $CaCl_2$—2.8 (g/kg diet); $KH_2 PO_4 \cdot H_2O$—4.5 (g/kg diet). The dietary energy was calculated as carbohydrate: 17.15 MJ/kg, protein: 23.64 MJ/kg, lipid: 39.54 MJ/kg.

### 2.3. Ammonia Nitrogen Challenge Assay

After being fed with experimental diets for eight weeks, a challenge assay was conducted on the fish. Ammonium chloride (NH$_4$Cl) was added into the aquaculture water at the concentration of 1 g/L. At six minutes, all the fish showed either manic agitation or lack of apparent consciousness; based on our pre-experiment, two fish from each tank were selected. One was sampled as the challenge assay sample, and the other was placed in water with no ammonia poison to recover. It was sampled after recovery from lack of apparent consciousness. Recovery time was about 5 min.

### 2.4. Sampling and Analysis

The body length and weight of the fish were measured at the beginning and the end of the feeding experiment. Then the specific growth rate, weight gain percentage, feed coefficient and condition factor were calculated. The calculation formulas are as follows:

Weight gain percentage (%) = (final average weight—initial average weight)/initial average weight × 100

Specific growth rate (%/d) = (Ln final average weight—Ln initial average weight)/breeding test days × 100

Condition factor = body weight/body length$^3$ × 100

Feed coefficient = feed weight/fish weight gain

In the formula, the unit of body length is cm, the unit of final average weight and initial average weight is g, and the unit of breeding test days is d.

At the two sampling times, fish blood was taken from the tail vein with a 1-mL syringe and was left to clot at room temperature for 2 h, then separated by centrifugation (3000× $g$, 10 min). Serum was collected and kept at −80°C until analysis. Serum was used to detect enzyme activity. Total cholesterol (TC) was measured by the CHOD-PAP enzymatic-colorimetric method. Triglycerides (TG) were determined enzymatically by the Triglyceride method [39]. Albumin (ALB) was analyzed according to the method described by Doumas et al., (1971) [40]. Aspartate aminotransferase (AST/GOT) and alanine aminotransferase (ALT/GPT) were evaluated using a colorimetric method according to Reitman and Frankel (1957) [41]. Lysozyme (LZM) activity was determined using a turbidimetric assay as in Ref. [42].

In addition, the liver, head kidney, intestine, gill and spleen of fish were sampled and put into centrifuge tubes separately, followed by immediate storage at −80 °C until analyzed. A nine-fold volume of 0.86% saline solution equal to weight was added in tissue samples. The tissue was then homogenized using the handheld homogenizer (Gloucester, Prima PB100, England) on ice. The homogenate was centrifuged for 20 min at 2500× $g$, and the supernatant was used for the further analysis. Amylase, Lipase and Trypsin were analyzed according to the method described by Reimer (1982) [43]. Na$^+$K$^+$ adenosinetriphosphatase (Na$^+$/K$^+$–ATPase) activity determination was performed according to Ay et al., (1999) [44]. Creatine kinase (CK) activity was determined using the method of Weng et al., (2002) [45]. γ-glutamyl transpeptidase (γ-GT) activities were measured using the method of Rosalki et al., (1970) [46].

All assays were determined using commercial assay kits (Nanjing Jiancheng Bioengineering Research Institute, Nanjing, China) according to the manufacturer's instruction.

### 2.5. Gene Expression

All fish tissue samples were taken to extract total RNA, according to the manufacturer's protocol for Trizol reagent (Invitrogen, Thermo Fisher Scientific Co., Ltd., Shanghai, China). A micro ultraviolet spectrophotometer (ND5000, Bioteke Corporation, Beijing, China) and agarose gel electrophoresis were used to test the concentration and purity of RNA. RNA was considered valid when OD260/280 (Optical density) was in the range of 1.9~2.1. 1 µg total RNA, drawn off for the reverse transcription process following

the manufacturer's instruction for One-Step gDNA Removal and cDNA Synthesis SuperMix (EasyScript, Beijing TransGen Biotech Co., Ltd., Beijing, China).

Quantitative real-time PCR analysis was run by Real-Time PCR System (Q1000, Hangzhou LongGene Scientific Instruments Co., Ltd., Hangzhou, China). The thermal cycling program was performed as follows: initial denaturation at 95 °C for 15 min, 40 cycles of denaturation at 95 °C for 10 s, annealing at 60 °C for 20 s, extension at 72 °C for 30 s. All PCR amplifications were performed in triplicate on 20 μL of reaction mixture containing 10 μL of 2×RealUniversal PreMix, 0.6 μL (10 μmol/L) of each primer, 2 μL of template cDNA, and 6.8 μL of RNase-free ddH2O. EF1 $\alpha$ and β-actin were used as the reference genes. A dissociation analysis was performed to determine the absence of nonspecific products at the end of each PCR reaction. A single peak was seen on the melt curve analysis, which indicated a single PCR product. A standard curve was established for 10-fold serial dilutions of cDNA for each primer pair. PCR reaction efficiency reached the range of 90–110%. The information on primers used for the analysis is provided in Table 2. The relative expression of HSP70 and HSP90 were determined in liver, head kidney, gill and spleen tissues. The relative expression of NF-κB1, TNF-$\alpha$, IL-1β, IL-8, TGF-β1, C3, C4, IgT, Hepc, IFN-$\gamma$, and Mx were determined in intestine tissue.

**Table 2.** Primer sequence table.

| Gene Abbreviation | Primer Sequence (5'-3') | Amplicon Size (bp) | Accession Number |
|---|---|---|---|
| NF-κB1 | F:CACAGACAGTTCGCCATCG | 185 | XM_022761336.1 |
| | R:AGCGTCTTCTGCCTCTTCC | | |
| [1] TNF-$\alpha$ | F:GAAAACGCTTCATGCCTCTC | 212 | XM_022746377.1 |
| | R:GTTGGTTTCCGTCCACAGTT | | |
| [1] IL-1β | F:TGATGGAGAACATGGTGGAA | 205 | XM_022753745.1 |
| | R:GTCGACATGGTCAGATGCAC | | |
| [1] IL-8 | F:GAAGCCTGGGAGTAGAGCTG | 164 | XM_022758559.1 |
| | R:GGGGTCTAGGCAGACCTCTT | | |
| TGF-β1 | F:CGGAGCTGCGGATGTTAA | 111 | XM_022738547.1 |
| | R:TGGTGATGAAGCGGGAAG | | |
| C3 | F:CATCGTTCCGCATCATAGC | 81 | XM_022755728 |
| | R:AGTCCTTGACATCCACCCA | | XM_022755434 |
| C4 | F:ACATCGCAATGGAGGAGAAC | 170 | XM_022768450.01 |
| | R:CAGTCCCGTGATAGGCTTTA | | |
| [2] IgT | F:TGGACCAGTCGCCATCTGAG | 196 | XM_022756471.1 |
| | R:GGGAAACGGCTTTGAAAGGA | | |
| [2] Hepc | F:GATGATGCCGAATCCCGTCAGG | 99 | XM_022764299.1 |
| | R:CAGAAACCGCAGCCCTTGTTGGC | | |
| IFN-$\gamma$ | F:TCTGTCTGACCCTCTGGTTTTC | 136 | LC146385.1 |
| | R:AAGATGGGCTTCCCGCTA | | |
| Mx | F:GACTTGGCTCTACCTGCTATCG | 177 | XM_022744797.1 |
| | R:GCTTATCTTTCCGTACCACTCC | | |
| HSP70 | F:CACGTATTCTTGCGTTGGG | 146 | XM_022741879.1 |
| | R:TCATGGCGACCTGGTTCT | | |
| HSP90 | F:GGCTACATGGCCGCTAAA | 187 | XM_022764360.01 |
| | R:TGCGATTGGAGTGGGTCTG | | |
| [3] EF1 $\alpha$ | F:ATCGTTGCCGCTGGTGTT | 134 | XM_022744048.1 |
| | R:TCGGTGGAGTCCATCTTGTT | | |
| [1,3] β-actin | F:TCTGGTGGGGCAATGATCTTGATCTT | 212 | XM_022757055.1 |
| | R:CCTTCCTTCCTCGGTATGGAGTCC | | |

[1] Primers of TNF-$\alpha$, IL-1β, IL-8 and β-Actin were based on the study of [47]. [2] Primers of IgT and Hep were based on the study of [48]. [3] EF1-$\alpha$ and β-actin were used as reference genes.

*2.6. Statistical Analyses*

The relative quantity of gene expression was calculated according to the $2^{-\Delta\Delta Ct}$ method [49]. The results were analyzed using the SPSS 19.0 statistical software packages. All data are expressed as the standard deviation (mean ± SD). Comparisons between different groups were conducted by one-way ANOVA and Tukey's test. Significant difference was considered at $p < 0.05$.

## 3. Results

Growth performance indicators (specific growth rate, weight gain percentage, feed coefficient and condition factor) after the feeding experiment are shown in Table 3. The condition factor of *Seriola dumerili* in the 0.01% treatment group was significantly higher than in the other two groups ($p < 0.05$), while there was no significant difference between the control group and the 0.02% treatment group ($p > 0.05$). The feed coefficient of *Seriola dumerili* in the control group was significantly higher than that in all treatment groups ($p < 0.05$), and there was no significant difference between treatment groups ($p > 0.05$). Moreover, different levels of dietary curcumin supplementation had no significant effects on weight gain percentage or specific growth rate of *Seriola dumerili* ($p > 0.05$).

**Table 3.** Growth performance of *Seriola dumerili* at different levels of curcumin in feed.

| Growth Performance | Group | | |
|---|---|---|---|
| | Control Group | 0.01% Treatment Group | 0.02% Treatment Group |
| Initial weight (g) | 149.02± 12.59 | 153.51 ± 4.09 | 151.78± 3.85 |
| Final weight (g) | 263.59 ± 83.29 | 327.38 ± 48.38 | 306.55 ± 23.95 |
| Weight gain percentage (%) | 73.89 ± 53.73 | 116.44 ± 34.07 | 102.30 ± 12.83 |
| Specific growth rate (%/d) | 1.26 ± 0.09 | 1.34 ± 0.25 | 1.25 ± 0.12 |
| Feed coefficient | 3.45 ± 0.33 [a] | 2.21 ± 0.21 [b] | 2.30 ± 0.31 [b] |
| Condition factor | 2.36 ± 0.05 [b] | 2.48 ± 0.06 [a] | 2.27 ± 0.05 [b] |

Data are presented as mean ± SD. In the same row, the same lowercase letters on the right side of the data indicated no significant difference ($p > 0.05$). Different lowercase letters indicated a significant difference ($p < 0.05$).

Digestive enzyme activities in the intestine of greater amberjack are shown in Figure 1. Only the 0.02% CUR group exhibited a significant difference with the control as to AMS activity ($p < 0.05$). Regarding LPS activity, two treatment groups had a similar difference with the control. There was no remarkable difference among all the groups regarding TPS activity ($p > 0.05$).

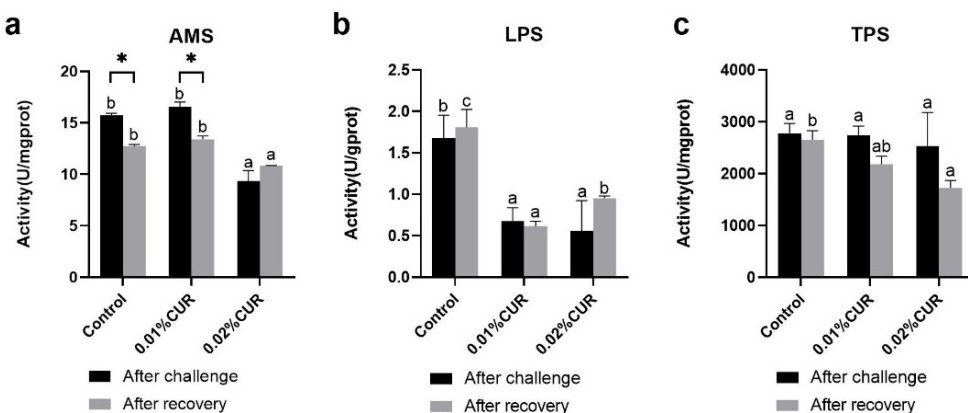

**Figure 1.** (**a**) Amylase (AMS), (**b**) Lipase (LPS), and (**c**) Trypsin (TPS) activities of greater amberjack fed with dietary curcumin supplementation. (Different letters above bars and asterisk sign (*) indicate significant differences at the 0.05 level).

The activity of Na$^+$/K$^+$–ATPase, $\gamma$-GT, and CK in intestinal tissue was also determined. The results are described in Figure 2. Na$^+$/K$^+$–ATPase activity in 0.02% CUR group is apparently lower than in the other two groups after the ammonia nitrogen challenge. An evident lower level was found in $\gamma$-GT activity in the 0.01% CUR group. The CK activity of both treatment groups was significantly higher than in the control.

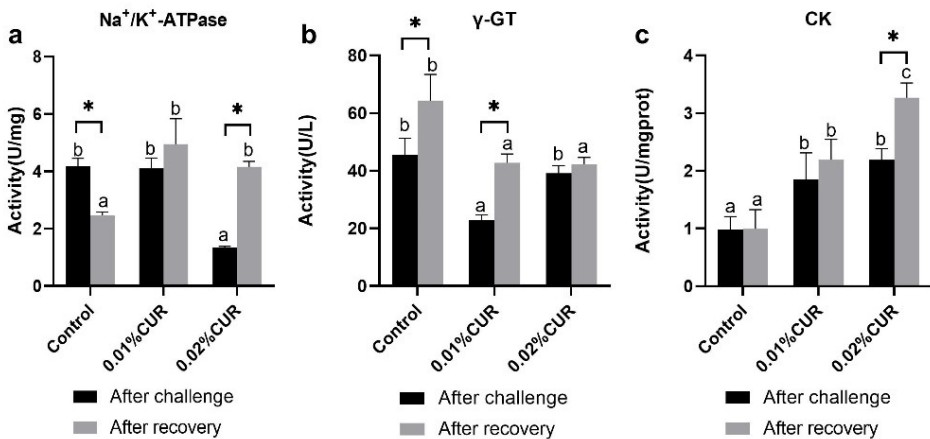

**Figure 2.** (**a**) Na$^+$K$^+$ adenosinetriphosphatase (Na$^+$K$^+$–ATPase), (**b**) $\gamma$-Glutamyl transferase ($\gamma$-GT), (**c**) creatine kinase (CK) activities of greater amberjack fed with dietary curcumin supplementation. (Different letters above bars and asterisk sign (*) indicate significant differences at the 0.05 level).

LZM and ALB (Figure 3), AST and ALT (Figure 4) activity, as well as TG and TC (Figure 5) content of serum were determined after the ammonia nitrogen challenge. According to Figure 3, LZM activity of both two treatments groups was higher than that of the control, and there was no significant difference between them. In addition, ALB content of serum showed no difference among groups. The variations in AST and ALT activity were quite different. In particular, no sharp distinction was seen in AST activity among all groups. With respect to ALT activity, the 0.02% CUR group compared to other groups was lower after the ammonia nitrogen challenge. In addition, different change characteristics were also revealed in the TG and TC content of serum. The TG content of all treatment groups was obviously higher than that in the control. By contrast, the TC contents of both treatment groups were lower.

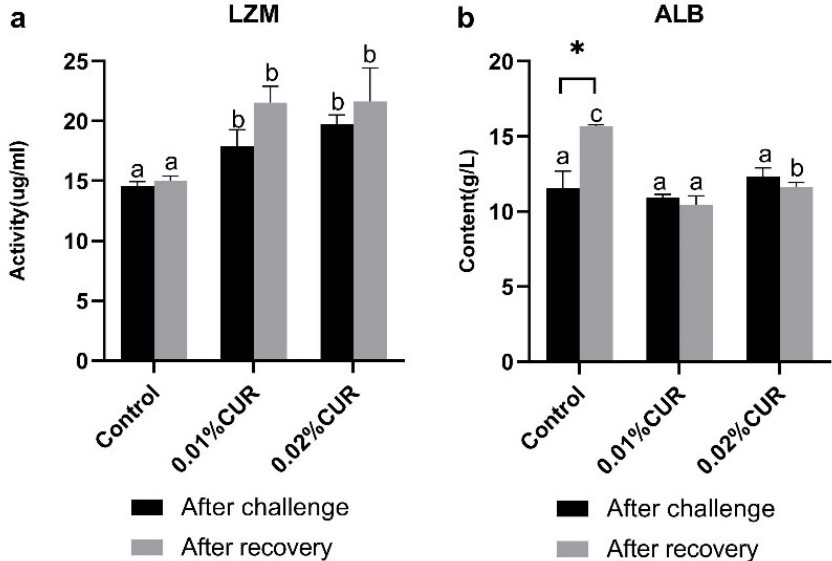

**Figure 3.** (**a**) Lysozyme (LZM) activity and (**b**) albumin (ALB) content of greater amberjack fed with dietary curcumin supplementation. (Different letters above bars and asterisk sign (*) indicate significant differences at the 0.05 level).

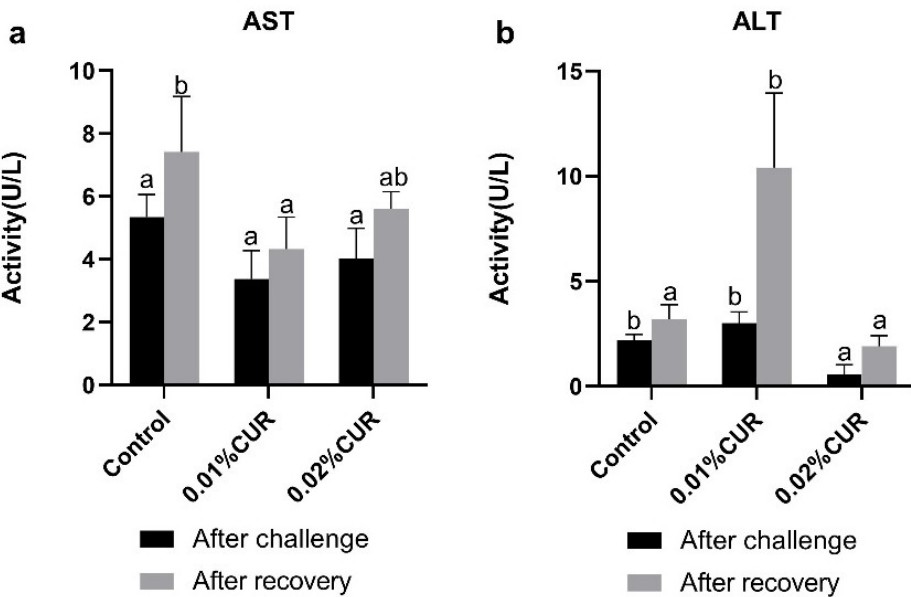

**Figure 4.** (**a**) Aspartate aminotransferase (AST) and (**b**) alanine aminotransferase (ALT) activities of greater amberjack fed with dietary curcumin supplementation. (Different letters above bars indicate significant differences at the 0.05 level).

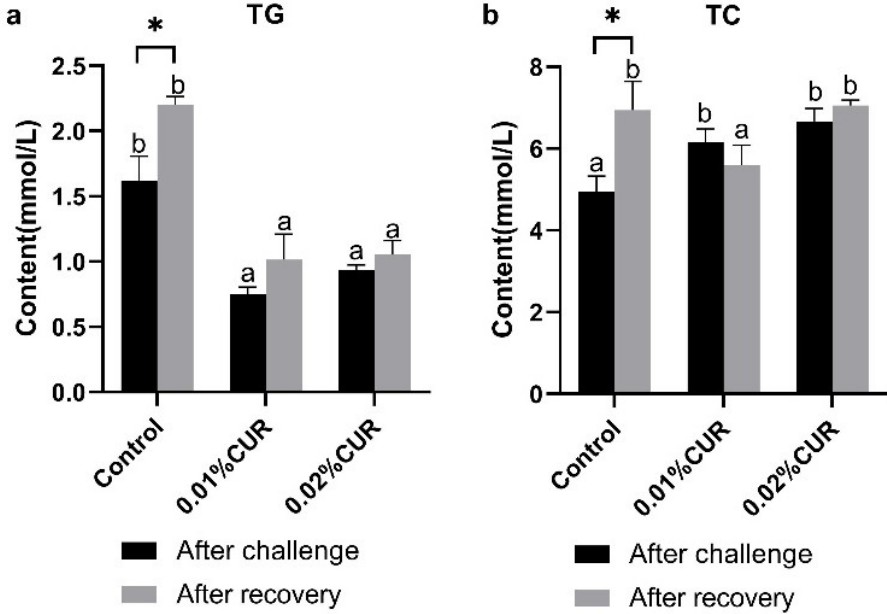

**Figure 5.** (**a**) Triglyceride (TG) and (**b**) total cholesterol (TC) content of greater amberjack fed with dietary curcumin supplementation after ammonia nitrogen challenge assay. (Different letters above bars and asterisk sign (*) indicate significant differences at the 0.05 level).

After recovery from apparent lack of consciousness, fish were sampled and the same in vivo indexes were determined as in the ammonia nitrogen challenge experiment. The activity of three digestive enzymes all showed a significant decline phenomenon compared with the control. The declines in AMS and TPS activity were seen in the 0.02% CUR group and the decline in LPS activity was found in both two curcumin-added groups. The levels of Na+/K+−ATPase and CK activity of the intestine in the two treatment groups after recovery were significantly higher than in the control. γ-GT activity in the two treatment groups was obviously lower than that of the control. Moreover, in serum, LZM activity in the two treatment groups was still maintained at a higher level than in

the control. The ALB content of the control reached the highest level after recovery. Figure 4 provides the results of the AST and ALT activity. The lowest AST activity level appeared in the 0.01% CUR group, while in contrast the ALT activity in the 0.01% CUR group was the highest. In addition, the TG in 0.01% CUR and 0.02% CUR groups were both significantly lower than in the control according to Figure 5. Regarding the TC content, the same significant difference was seen only in 0.01% CUR.

Heat shock protein genes, containing HSP70 and HSP90, were quantified in liver (a), kidney (b), spleen (c) and gill (d) tissues of greater amberjack after the challenge assay (Figures 6 and 7). Regarding the HSP70 gene, the relative expression of the HSP70 gene of experimental groups in the liver was significantly lower than that of the control after the treatment challenge assay. In contrast, the relative expression of the HSP70 gene in the kidney and spleen of two curcumin-added groups were both higher than that of the control. In the gill, only the relative expression of the HSP70 gene of the 0.02% CUR group was at the highest level.

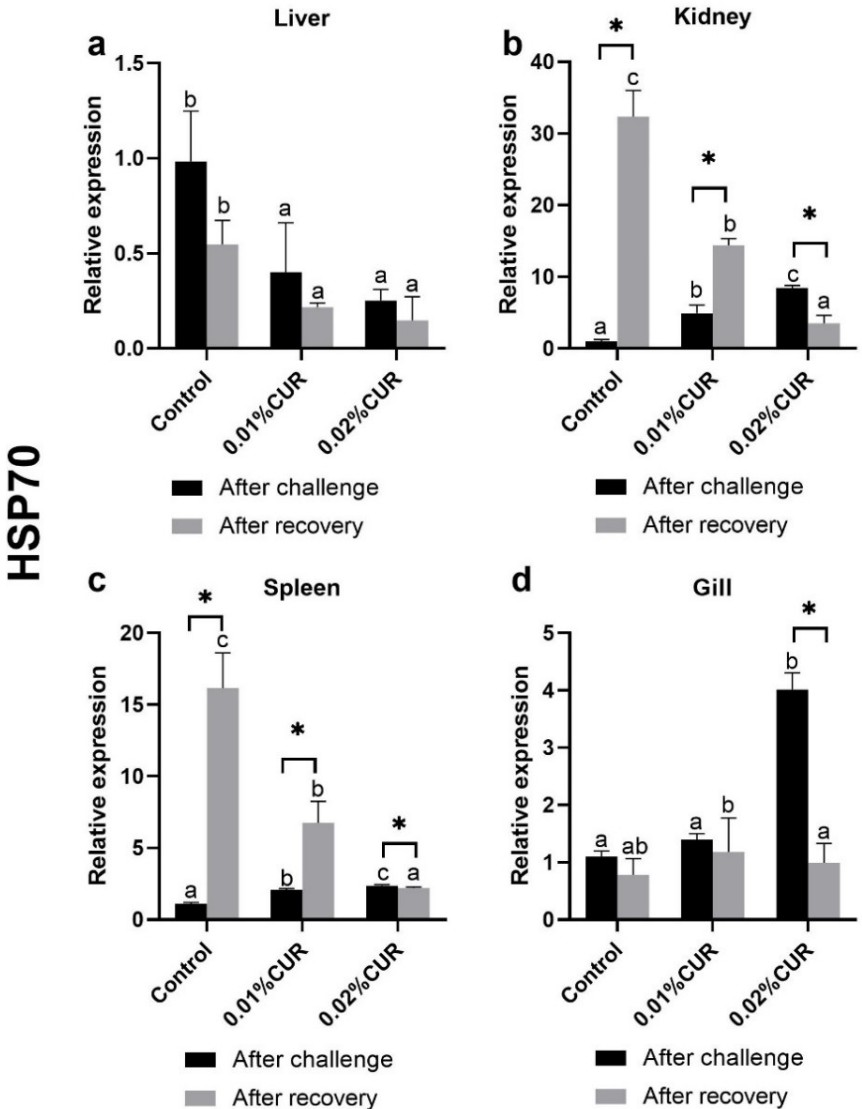

**Figure 6.** The relative expression of heat shock protein (HSP70) genes in (**a**) liver, (**b**) kidney, (**c**) spleen and (**d**) gill tissues of greater amberjack fed with dietary curcumin supplementation. (Different letters above bars and asterisk sign (*) indicate significant differences at the 0.05 level).

The rules of the relative expression of HSP90 gene in each tissue were similar with those of HSP70. In kidney and spleen tissues, the relative expression of the HSP90 gene in

both the two curcumin-added groups was significantly higher than that of the control, but the HSP90 expression level of the 0.02% CUR group was significantly lower than that of the 0.01% CUR group. The relative expressions of the HSP90 gene in liver were higher with the increase in the amount of added curcumin. Moreover, the change situation of the relative expressions in gills showed a huge difference between the two heat shock protein genes. The relative expression of the HSP90 gene in gills of the 0.02% CUR group was the highest; however, in HSP70, the relative expression of the HSP90 gene of the control was the highest.

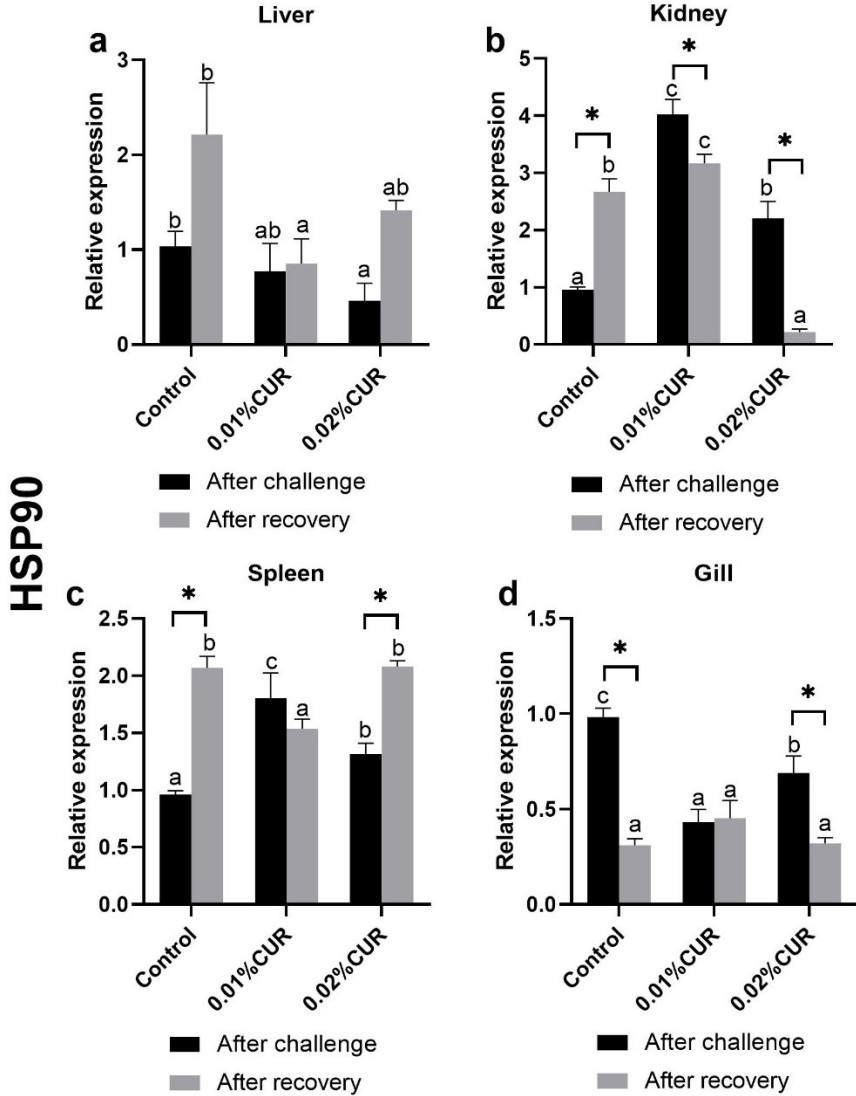

**Figure 7.** The relative expression of heat shock protein (HSP90) genes in (**a**) liver, (**b**) kidney, (**c**) spleen and (**d**) gill tissue of greater amberjack fed with dietary curcumin supplementation. (Different letters above bars and asterisk sign (*) indicate significant differences at the 0.05 level).

There have been some changes in the relative expression of the HSP70 and HSP90 genes in four tissues after recovery compared to the state after the ammonia nitrogen challenge (Figures 6 and 7). As for the HSP70 gene, its relative expression in liver, kidney and spleen was obviously less than that of the control group, but in gill, its relative expression had no difference among all the groups. It was noted that the situation of the relative expression of HSP90 in gill was also quite similar with that of HSP70; experimental groups also showed no difference with the control group. Furthermore, the relative expression of HSP90 in liver was still kept at a lower level than the control group

after recovery. However, the relative expression of HSP90 in kidney and spleen fell into a different situation. In spleen in particular, only the relative expression of HSP90 in the 0.01% CUR group different from other groups. In kidney, however, a higher value was found in the 0.01% CUR group and a lower value in the 0.02% CUR group compared with the control.

The relative expression of cytokine IL8, IL1β, TNF-α, IFN-γ and TGF-β1 genes in intestines of greater amberjack after the challenge assay are given in Figure 8. A lower level occurred both in the relative expression of the IL8 and TGF-β1 genes in two curcumin-added groups compared with the control. Regarding the TNF-α gene, only the high curcumin-added group showed the lower level. However, IL1β gene and IFN-γ gene relative expressions of the curcumin-added group were all significantly higher than that of the control. The difference is that the highest level of the IL1β gene occurred in the 0.02% CUR group, and that of the IFN-γ gene occurred in the 0.01% CUR group.

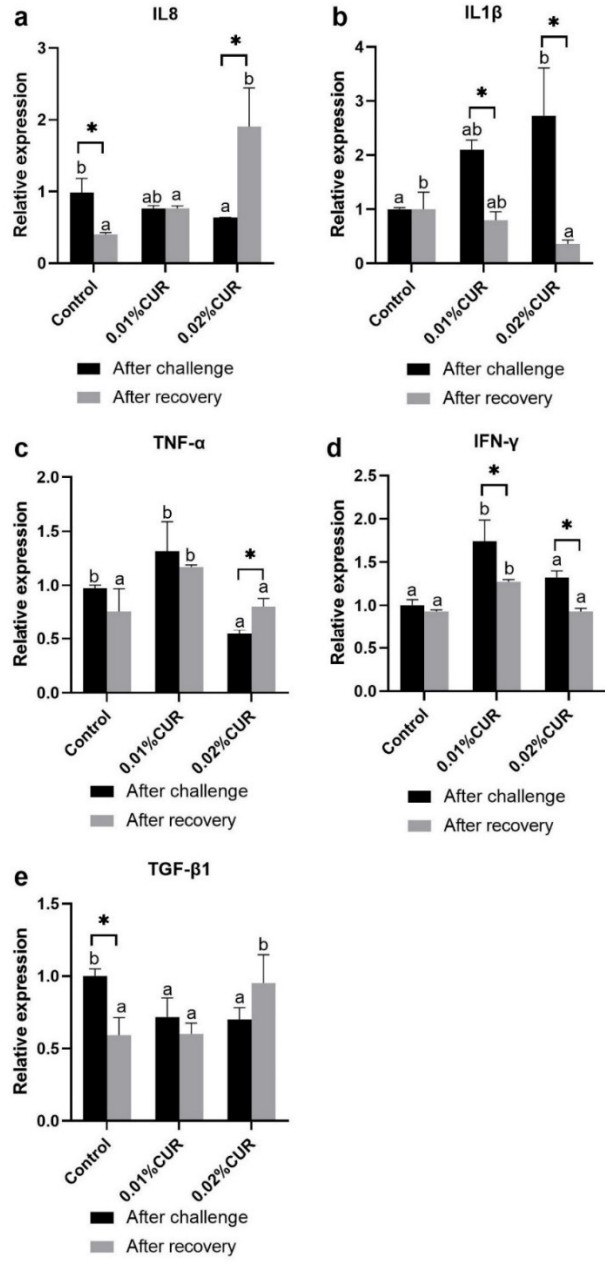

**Figure 8.** The relative expression of (**a**) cytokine IL8, (**b**) IL1β, (**c**) TNF-α, (**d**) IFN-γ and (**e**) TGF-β1 genes in intestine of greater amberjack fed with dietary curcumin supplementation. (Different letters above bars and asterisk sign (*) indicate significant differences at the 0.05 level).

The relative expression of other genes (C3, C4, NF-κB1, Mx, Hepc and lgT) after challenge are analyzed in Figure 9. C3 and lgT genes were both found to have a higher relative expression in the 0.01% CUR group and a lower relative expression in the 0.02% CUR group. The relative expression of C4 and Hepc genes showed the same phenomenon; in both there appeared a distinct lower value in the 0.02% CUR group compared with other groups. In addition, NF-κB1 and Mx genes were on the opposite sides. Treatment groups in particular were significantly higher than the control regarding the NF-κB1gene and lower regarding the Mx gene after the ammonia nitrogen challenge compared with the control.

The results of the relative expression of some cytokine genes and other related genes after recovery are also provided in Figures 8 and 9. As for cytokine genes, the highest relative expression of IL8 and TGF-β1 genes was witnessed in the 0.02% CUR group and the highest relative expression of TNF-$\alpha$ and IFN-$\gamma$ genes in the 0.01% CUR group instead. In addition, the IL1β gene was expressed the least in the 0.03% CUR group while the relative expression of IL1β in the 0.01% CUR group showed no significant difference to others.

The relative expression of the C3 gene in the 0.01% CUR group was remarkably lower than in other groups. On the contrary, the NF-κB1 gene's relative expression in the 0.01% CUR group was the highest among the groups. The relative expression of the C4 and Hepc genes had the same tendency: their level of expression in the 0.02% CUR group rose to a higher value compared with the control after recovery. Moreover, only the relative expression of the Mx gene in the 0.02% CUR group was significantly lower than in other groups, but both two curcumin-added groups witnessed a markedly lower value of the relative expression of the lgT gene than in the control.



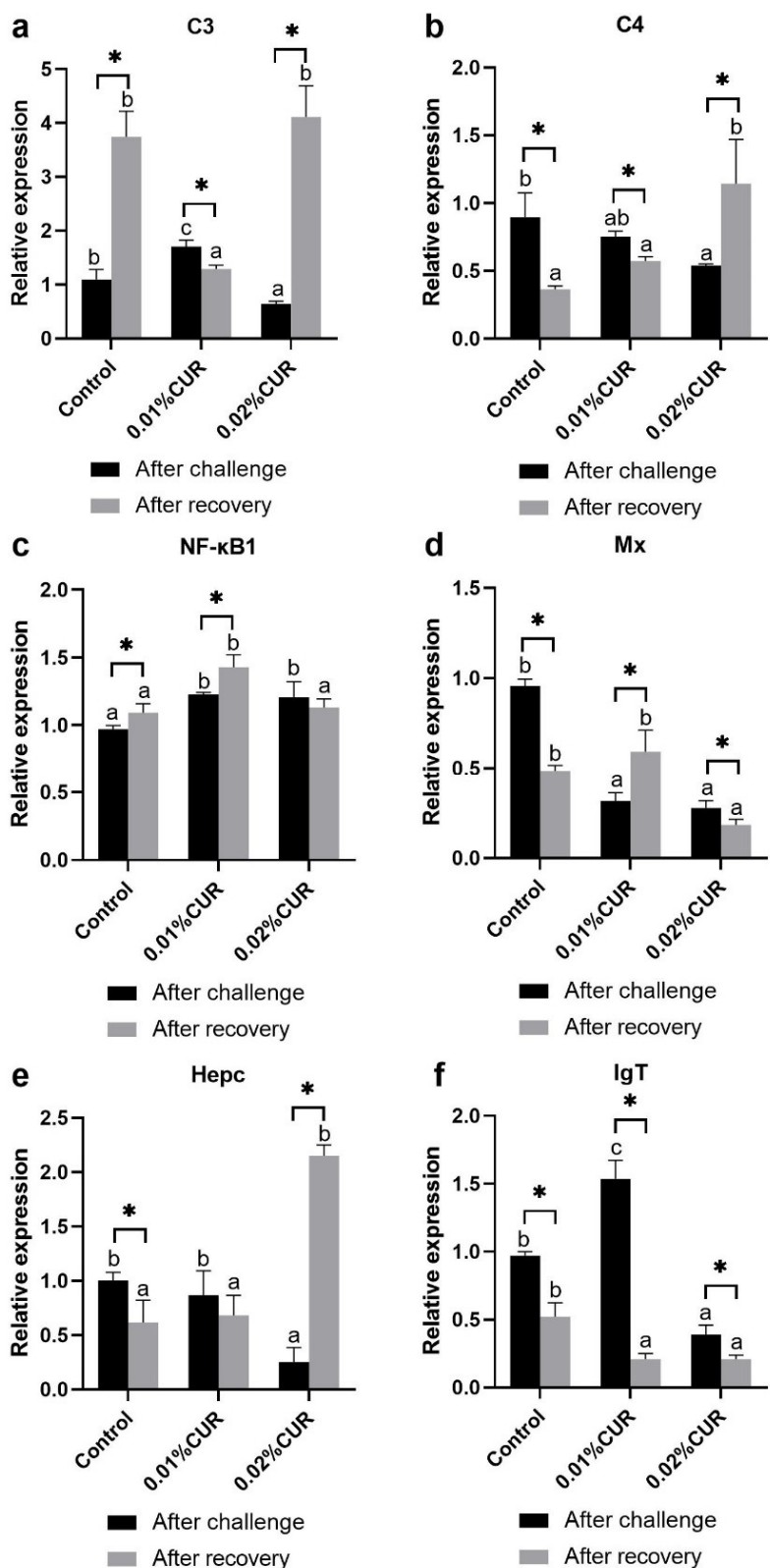

**Figure 9.** The relative expression of complements (**a**) (C3), (**b**) C4, (**c**) NF-κB1, (**d**) Mx, (**e**) Hepc and (**f**) lgT genes in intestine of greater amberjack fed with dietary curcumin supplementation. (Different letters above bars and asterisk sign (*) indicate significant differences at the 0.05 level).

## 4. Discussion

In order to determine whether the addition of curcumin in the diet has an effect on the resistance to external environmental stress of greater amberjack, or whether it effectively protects the in vivo multiple functions of fish and enhances the capacity of recovery from stress, we analyzed and confirmed the results from multiple perspectives, such as nutrition, organ protection and the immune system, etc.

The role of dietary curcumin supplementation in animal nutrition is our first concern, on account of the digestion and absorption ability of fish having direct and important links with final growth performance [50–52]. Digestive enzymes including amylase, lipase, and trypsin were determined after two phases of the experiment. Comparisons of digestive enzyme activity after acute ammonia nitrogen stress and recovery provided the evidence of the promotion of recovery of digestive functions owing to the dietary curcumin supplementation. In particular, the activity of amylase and lipase of greater amberjack fed with 0.02% curcumin additive was not only maintained at the previous level but also increased. In contrast, the counterparts of the control and the 0.01% additive amount group decreased together and showed no change in their relative relationship, indicating that the high content of the curcumin additive maintained and even strengthened the activity of amylase and trypsin during the recovery following stress. Similar findings occurred in other fish. For instance, in crucian carp (*Carassius auratus*), significant increases were found in trypsin and lipase activities in the intestine after supplemental doses of dietary curcumin at 5 g/kg [35]. In addition, Ref. [53] demonstrated the ascent of these two digestion enzymes in another fish, *Ctenopharyngodon idella*, along with a rise in amylase activity. However, in the present study, the same positive assistance was not seen in trypsin activity in the fish fed at both dosage rates of curcumin additive, which is inconsistent with previous studies. This contradiction may be due to an insufficient dose, or there may be another possibility that should be further investigation by experiment. As for absorption capacity, in our experiment the enhanced effect of curcumin supplementation on Na$^+$/K$^+$–ATPase, $\gamma$-GT and CK activity was more obvious. The recovery mechanism was started earlier, compared with the control, regardless of the additive amount; in other words, the stress recovery process was accelerated. The studies mentioned above on crucian carp and grass carp also witnessed a positive increase in these three enzymes related to nutrition absorption with additional curcumin supplementation in the fish diet. Furthermore, a great deal of research has provided more direct enhanced-growth data more intuitively (increases in WG, SGR, DWG, FCR and FE, etc.) [54–58]. Collectively, it can be concluded that dietary curcumin supplementation can not only increase growth in daily culture by improving digestion and absorption, but can also raise the performance in a stress environment.

The role of curcumin as an immunostimulant that stimulates the activity of the non-specific immune system in fish is also worthy of attention. The activity of lysozyme is an important marker for representing in vivo non-specific immunity of fish [59,60]. In this study, when there was no significant change in the control group, lysozyme activity increased in the fish fortified with both curcumin diets. The improving effect of curcumin on lysozyme activity was also reported by Leya et al. [61], who fed *Labeo rohita* fingerlings a diet with 0.5%, 1%, 1.5% and 2% curcumin for 60 days; a significant rise was observed in lysozyme activity from 15 days to the end. Furthermore, another study also showed an increase in the lysozyme activity of *Oreochromis niloticus* fed the 50 mg and 100 mg/kg curcumin diet, but not in all the groups [38]. That indicated that the benefits of the additive amount of curcumin did not improve with higher amounts; however, the proportion of 0.01% and 0.02% in our present study remained positive for lysozyme activity. Albumin (ALB) should be another important component of a non-specific immune system [62,63]. Ref. [64] studied the effect of a curcumin-supplemented diet in *Cirrhinus mrigala* and found that fishes fed a curcumin-enriched diet (1–1.5%) showed significantly higher serum albumin. Similarly, research into the effects of two kinds of Supplementary Materials on *Nile tilapia* immunity also witnessed a remarkable increase in the curcumin-

addition group [65]. However, the ALB of the two treatment groups showed no obvious change, while the ALB of the control pulled away from the state of no significant difference among the three groups. The results of the ALB were inconsistent with previous studies, and whether it was related to the amount of curcumin addition should be further explored.

Aspartate aminotransferase (AST) and alanine aminotransferase (ALT) are produced and released by hepatocytes, and they are considered biomarkers of liver damage injury [66–69]. As for the protective effect of curcumin on the liver, the involvement of AST and ALT has been reported in some research. A significantly lower level of ALT and AST activities were observed in *Megalobrama amblycephala* from the 60 mg/kg curcumin diet group [70]. In addition, a study on grass carp showed that ALT and AST activities decreased significantly on account of curcumin addition [71]. The protective effect of curcumin on the liver was also found in this study, and a greater reduction in serum AST or a quicker reduction in serum AST were observed. However, no similar trend occurred in ALT activity. This phenomenon illustrated that, at least at the curcumin level of this experiment, a protective effect of restricting ALT activity was not activated.

Curcumin has been proved to have lipid-lowering and anti-obesity effects on mammals [72–76]. Therefore, its function of improving lipid metabolism in fish was also what we wanted to understand, and triglyceride (TG) and total cholesterol (TC) were tested in this study for verification. The present study showed that TG and TC content were basically maintained at the original level or even showed a slight decrease when the control group increased significantly after recovery. Based on that, facilitation of lipid metabolism on greater amberjack by curcumin has been very clear. These results are consistent with the results of a study [36] in which TC, TG and LDL-c were observed in the curcumin treatment group.

Heat-shock proteins (HSPs) are a highly conserved protein family; commonly analyzed HSP70 and HSP90 are included. These can be induced by a variety of stressors and perform their protective roles. Many heat shock proteins have molecular chaperone activity to preserve the tertiary structure of other proteins. Therefore, the proteins are an important part of the cellular stress response and considered excellent markers of stress response [77–81]. The relative expressions of HSP70 and HSP90 in four tissues (liver, kidney, spleen and gill) were determined in this study. Regarding HSP70, through the comparison of stress state and recovery state, curcumin obtained an outstanding performance in the relative expressions of HSP70 in fish, but its effect on liver cannot be clarified from the data. Regarding HSP90, its situation was very different from that of HSP70. HSP90 in all tissues except gill remained in the state of stimulated stress and did not fully recover. Nonetheless, the positive effect of curcumin also can be found in liver, kidney and spleen tissues. It should be noted, however, that different additive amounts of curcumin resulted in great differences.

Cytokine is a class of small molecular proteins with a wide range of biological activities. The class mainly includes interleukins (ILs), tumor necrosis factors (TNFs), transforming growth factor-β family (TGF-β family) and interferon (IFN), etc. [82]. Cytokine also can be roughly divided into two types: pro-inflammatory cytokines and anti-inflammatory cytokines. Pro-inflammatory cytokines determined in this study were IL8, IL1β, TNF-$\alpha$ and IFN-$\gamma$, and an anti-inflammatory cytokine was TGF-β1. The function related to the anti-inflammation role of curcumin has been widely reported [83,84]. The study in rats reported that the expression of TNF-$\alpha$ and IL-6 in hypercholesterolemic rats decreased after a 200 mg/kg curcumin treatment. In fish, Ref. [85] investigated the protective effects of curcumin on liver-damaged *Cyprinus carpio*; the results indicated that 0.5% and 1.0% curcumin reduced the CCl4-induced damage in liver by inhibiting NF-kB, IL-1β, TNF-$\alpha$, and IL-12 expression. Regarding pro-inflammatory cytokines in the present study, curcumin showed only its inhibition capacity on IL1β and IFN-$\gamma$ for their sharper declines during recovery. In addition, the sole anti-inflammatory cytokine TGF-β1 we determined to be obviously improved in the high curcumin-dosage group. Transcription

factor nuclear factor kappa B (NF-κB) has a central role in promoting the process of inflammation and cancer, and cytokine expressions can be activated via this signaling pathway [86,87]. Curcumin inhibits the expression of cytokine via blocking the NF-κB pathway. Ref. [88] provided the evidence with the study on *Oncorhynchus mykiss*; NF-κB expression declined dramatically after curcumin was added to the diet. Another test on *Cyprinus carpio* showed an attenuation effect on the expression of NF-κB caused by Pb exposure [89]. The result of this study was similar with theirs. The expression of NF-κB1 decreased to a higher degree in fish fed with 0.02% dietary curcumin, thus avoiding the continuation of inflammation response more quickly.

Some other immune-related genes were determined in this study; both effector molecules related to the innate immune system and the adaptive immune system were included (C3, C4, Hepc, Mx and lgT). The complement system comprises more than 35 proteins and plays an important role in the innate immune system of fish. It can exert positive effects on phagocytosis, antigen-antibody aggregate clearance, inflammatory reactions, and antibody production [90–93]. Antimicrobial peptides (AMPs) are also a vital component of the innate immune system which cannot be ignored. Hepcidin (Hepc), as a member of AMPs which are rich in cysteine, also performs well in antimicrobial activity. Coupled with its specific iron regulation function, it is of great importance to organism [94–96]. In addition, proteins encoded by the Mx gene are high-molecular-weight GTPases induced by interferon; they have strong intrinsic GTP hydrolysis, broad-spectrum antiviral activity and can inhibit a variety of negative-strand RNA viruses [97,98]. Immunoglobulins (Igs) play a key role in the adaptive immune system to help the organism against pathogens. Immunoglobulin T (lgT) was the last found among all immunoglobulins in teleost fish. It has been proved to have a more specific focus on gut mucosal immunity compared to other immunoglobulins (IgM, IgD, IgG, IgE, and IgA) [99–101]. The expression of these genes was sufficient to reflect the status of many important parts of the fish immune system, which enabled us to glimpse the truth of immune changes. In the present study, no striking enhancement was found in the restorative ability of C4, Hepc and Mx under both proportions of dietary curcumin treatments. However, 0.01% dietary accelerated the recovery pace of C3 and lgT after stress. Although positive results were not acquired for all the immune-related genes, the possibility that curcumin improves the immune functions of greater amberjack is assured, and more exciting results can be obtained on more indexes after more investigations into additive schemes.

Recent research into dietary curcumin supplementation on more fish species than our present study covers further broadens its application. For example, dietary curcumin proved to promote gilthead seabream larvae digestive capacity and modulate the oxidative status [102], and it was also found to work at the post-larval stage [103]. Dietary curcumin was also found to relieve the lipopolysaccharide (LPS)-induced and deltamethrin (DEL)-induced stress response (oxidative stress, inflammation and cell apoptosis) in *Channa argus* via Nrf2 and NF-κB signaling pathways [104,105]. With more related studies combined with the results of our research, it can be predicted that dietary curcumin supplementation will play an important role in the healthcare and maintenance of robustness of fish.

## 5. Conclusions

In brief, the application prospects for dietary curcumin supplementation in greater amberjack can be confirmed based on the results of this study. Curcumin has shown enhanced capability of protective effect and stress recovery in nutritional function, anti-inflammation, immune system, disease resistance, tissue protection, etc. These results are consistent with previous studies. In addition, our research shows that the relationship between the effect and the additive amount was not linear, and that a greater amount of additive did not necessarily lead to a better effect; therefore, a wider range of additive-amount experiments should be pursued. Eventually, the curcumin effect curve in greater

amberjack will be obtained through these works, in order to provide a better reference for future practical applications.

**Supplementary Materials:** The following supporting information can be downloaded at: https://www.mdpi.com/article/10.3390/jmse10111796/s1.

**Author Contributions:** Z.M. and J.H. (Jiawei Hong) conceptualization of the experiment; J.H. (Jiawei Hong) and Z.F. methodology and performance of the experiment; J.H. (Jing Hu) performed the statistical analysis; J.H. (Jiawei Hong) and Z.F. writing—original draft preparation; S.Z. writing, review and editing; G.Y. and Z.M. supervised the final version of the paper. All authors have read and agreed to the published version of the manuscript.

**Funding:** This study was supported by Central Public-interest Scientific Institution Basal Research Fund, CAFS (NO. 2020TD55), Guangxi Innovation Driven Development Special Fund Project [Guike AA18242031].

**Institutional Review Board Statement:** This study was carried out in strict accordance with the recommendation in the Animal Welfare Committee of Chinese Academy of Fishery Sciences. No protected species were used during the experiment.

**Informed Consent Statement:** Not applicable (Present study did not involve humans). Written informed consent has been obtained from all the patients to publish this paper.

**Data Availability Statement:** The data presented in this study are available on request from the corresponding author.

**Conflicts of Interest:** The authors declare no conflict of interest.

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
