# Peer review of "Dietary Curcumin Supplementation Enhanced Ammonia Nitrogen Stress Tolerance in Greater Amberjack (Seriola dumerili): Growth, Serum Biochemistry and Expression of Stress-Related Genes"

_jmse, doi:10.3390/jmse10111796_

Round 1

Reviewer 1 Report

I believe there is a mistake in tables 1 and 3, specifically in relation to the amounts of crude protein and dry matter in the first table, and the weight gain rate values ​​in the third table.

Author Response

Response to Reviewer 1 Comment

Point 1: I believe there is a mistake in tables 1 and 3, specifically in relation to the amounts of crude protein and dry matter in the first table, and the weight gain rate values ​​in the third table.

Response 1: I'm sorry for my mistakes. All the amounts and values have been checked again and the mistakes in table 1 and 3 have been revised in original place.

Reviewer 2 Report

I have read the manuscript that describes the efficiency of curcumin in amelioration of stress induced by ammonia nitrogen toxicity. I have detected some flaws and few suggestions needs to be incorporated. The English language needs major improvement 

line 23-24. rephrase the line "The expression of HSP70 and HSP90 was also have no 23 evident stimulate of in liver, kidney and spleen tissues"

Introduction needs much deeper literature survey of the impact of curcumin on the different physiological aspects of  cultured aquatic animals. line33-50  is a general background of the sector which should be avoided. 

Table-1: The  data presented is questionable, how is it possible to feed a fish with that low amount of crude protein and  crude lipid in diet.

mRNA expression in which you are showing  relative change with respect to control, how the control  is showing that much value because it should have a constant value of 1. Are you using something else as control or some other method? Figure 9b and 9d show a lesser value of control than 1. And after challenge the control value of the relative expression is not constant.

Author Response

Response to Reviewer 2 Comments

Point 1: line 23-24. rephrase the line "The expression of HSP70 and HSP90 was also have no 23 evident stimulate of in liver, kidney and spleen tissues"

Response 1: The sentence has been rephrased into ‘HSP70 and HSP90 genes weren’t evidently stimulated to express in liver, kidney and spleen tissues.’[line23-24]

Point 2: Introduction needs much deeper literature survey of the impact of curcumin on the different physiological aspects of cultured aquatic animals. line33-50 is a general background of the sector which should be avoided. 

Response 2: Thank you so much for your advices about introduction writing. The description of background has been revised [line33-46] and literature survey of the impact of curcumin on the different physiological aspects of cultured aquatic animals has been added[line 68-79].

Point 3: The data presented is questionable, how is it possible to feed a fish with that low amount of crude protein and crude lipid in diet.

Response 3: I'm so sorry for my mistakes here. All the amounts and values have been checked again and the mistakes in table 1 have been revised in original place.

 Point 4: mRNA expression in which you are showing relative change with respect to control, how the control is showing that much value because it should have a constant value of 1. Are you using something else as control or some other method? Figure 9b and 9d show a lesser value of control than 1. And after challenge the control value of the relative expression is not constant.

Response 4: We use the control after challenge to calculate the relative expression values of all other groups (including after challenge and after recovery).

Reviewer 3 Report

The study deals with an interesting topic and the number of parameters is enough to ensure publication as a full-length paper, the ms can be considered after some revisions

1- the authors should have a closer look at the updated literature on curcumin in aquaculture and involve in the introduction and discussion sections

2- Based on what inclusion levels were selected?

3-Provide reference for the challenge

4- Methods are not described properly and just referred to kits

5- some bars have no error bar which is not acceptable, and must be provided

Author Response

Response to Reviewer 3 Comments

Point 1: the authors should have a closer look at the updated literature on curcumin in aquaculture and involve in the introduction and discussion sections

Response1: literatures on curcumin in aquaculture were added in the introduction and discussion sections. [line68-79, 482-490]

Point 2: Based on what inclusion levels were selected?

Response2: We refer to previous studies. The most appropriate curcumin supplementation level was estimated to be 312.27 mg/kg in the diet of on-growing grass carp (Ctenopharyngodon idella) (Li et al., 2020). The appropriate dietary curcumin supplementation could enhance the growth (especially 60 and 120 mg kg-1 curcumin per feed) of common carp, Cyprinus carpio (Zhang et al., 2021). In addition, combined with preliminary experiments, the final values of curcumin added to Seriola dumerili feeds were 0, 100 and 200 mg/kg. Another published study by our team found that such a curcumin addition regime could result in significant differences in the growth, intestinal digestive enzymes and structure of greater amberjack.

reference

Li, G., Zhou, X., Jiang, W., Wu, P., Liu, Y., Jiang, J., et al. (2020). Dietary curcumin supplementation enhanced growth performance, intestinal digestion, and absorption and amino acid transportation abilities in on-growing grass carp (Ctenopharyngodon idella). AQUACULTURE RESEARCH 51(12), 4863-4873. doi: 10.1111/are.14777.

Yang, J., Hong, J., Fu, Z., & Ma, Z. (2022). Effects of Dietary Curcumin on Growth and Digestive Physiology of Seriola dumerili. Front. Mar. Sci, 9, 862379.

Zhang, Y., Song, L., Guo, H., Wu, J., Wang, X., and Yao, F. (2021). Effects of Curcumin on Growth and Liver protection in Common Carp, Cyprinus carpio. Pakistan Journal of Zoology 53(4).

Point 3: Provide reference for the challenge

Response3: The level of challenge experiment was based on our pre-experiment. We chose 5 concentrations (0.1g/L, 0.3g/L, 0.5 g/L, 1g/L and 3g/L) of ammonium chloride to determine the suitable experimental concentration. Our results showed that it’s not effective at the concentration of 0.1g/l and 0.3g/L. It came into effect above 15 min when added 0.5 g/L ammonium chloride, and not all fish. The concentration of 3g/L was so fast to make fish fell into manic agitation or faint and resulted in some death. However, 1g/L made fish fell into faint at about 6 min, and resulted in some death at 8 min. Therefore, we chose the concentration of 1g/L challenge for 6 min to do the challenge experiment.

Point 4: Methods are not described properly and just referred to kits

Response4: The description about methods has been revised. [line142-148, 154-161]

Point 5: some bars have no error bar which is not acceptable, and must be provided

Response5: I have checked all the error bars in all figure. Every bar has error bar. I don't know if it is the document display problem?

Reviewer 4 Report

The present study assesses the effect of dietary curcumin on ammonia stress tolerance in the greater amberjack. The aim of the study is interesting since promoting resistance to stocking is a goal in aquaculture, however, the manuscript needs some improvements. Firstly, English language must be deeply revised. In addition:

- Water temperature was described between 27 and 31.5ºC. The optimal temperature for the species has been described up to 26ºC, is the temperature range in the present study appropriate for the species? Or might be another stress factor?

- In the M&M section, it is stated that two fish were sacrified per tank, one after challenge and other after recovery, that means n = 3 per treatment and sampling point? If the initial density was 15 fish/tank results would be more robust with a higher n.

- How much time was needed for fish recovery? Was the same for the three dietary treatments? Recovery time must appear in the manuscript, if different time was needed between treatments should be indicated as a parameter.

- In the results, in the case of no statistical differences between treatments, please delete lowercase letters.

- Please, Indicate the correct abbreviation for each enzyme.

- Most of the discussion is based on differences between after challenge and after recovery levels, however, were all those comparisons statistically significant? Statistics in the figures indicate differences between treatments for each sampling point, but what about differences for the same treatment between sampling points? For example, authors discuss lipase and amylase activity levels before and after the recovery, but activities patterns are so similar. From the results in the control group, it seems that those enzymes are not affected by stress, but are by dietary treatments. Therefore, for all the study, to obtain clear conclusions, comparisons between pre- and post- recovery are needed.

Actualize references with recent bibliography on the topic:

Xavier, M.J. et al., 2022. Effects of dietary curcumin in growth performance, oxidative status and gut morphometry and function of gilthead seabream postlarvae. Aquaculture Reports 24, 101128. DOI: 10.1016/j.aqrep.2022.101128

Xavier, M.J. et al., 2021. Dietary curcumin promotes gilthead seabream larvae digestive capacity and modulates oxidative status. Animals 11(6), 1667. DOI: 10.3390/ani11061667

Author Response

Response to Reviewer 4 Comments

Point1: Water temperature was described between 27 and 31.5ºC. The optimal temperature for the species has been described up to 26ºC, is the temperature range in the present study appropriate for the species? Or might be another stress factor?

Response1: The optimal temperature for greater amberjack was determined by Fernández‐Montero et al. They concluded that the optimal temperature was 26ºC. However, they only chose three rearing temperatures (17, 22 and 26°C), there was no other temperature groups above 26°C. The optimal temperature they got was only among experimental three temperatures. In addition, the fish we used in our experiment were bred by ourselves and their parent fish were captured from the nature marine area in Lingshui, Hainan (where our institution is located). Nature sea water temperature they live and their rearing temperature was in this range.

References:

Fernández‐Montero, A., Caballero, M. J., Torrecillas, S., Tuset, V. M., Lombarte, A., Ginés, R. R., ... & Montero, D. (2018). Effect of temperature on growth performance of greater amberjack (Seriola dumerili Risso 1810) Juveniles. Aquaculture Research, 49(2), 908-918.

Fernández-Montero, A., Torrecillas, S., Tort, L., Ginés, R., Acosta, F., Izquierdo, M. S., & Montero, D. (2020). Stress response and skin mucus production of greater amberjack (Seriola dumerili) under different rearing conditions. Aquaculture, 520, 735005.

Point2: In the M&M section, it is stated that two fish were sacrified per tank, one after challenge and other after recovery, that means n = 3 per treatment and sampling point? If the initial density was 15 fish/tank results would be more robust with a higher n.

Response2: We sampled 3 fish per treatment after challenge and after recovery. Thank you for your advice about using higher n. Our institution started to investigate the culturing and breeding of greater amberjack recently years, it’s difficult to obtain parent fish and reproduce well so that the biological material is precious, we have tried our best to guarantee the scientificity under limited conditions.

Point3: How much time was needed for fish recovery? Was the same for the three dietary treatments? Recovery time must appear in the manuscript, if different time was needed between treatments should be indicated as a parameter.

Response3: It needed about 5min for fish recovery. The recovery time of three dietary treatments was the similar. The related description about recovery time must appear in the manuscript. [line125-126]

Point4: In the results, in the case of no statistical differences between treatments, please delete lowercase letters.

Response4: Thank you for your advices. All lowercase letters in the case of no statistical differences between treatments were deleted.

Point5: Please, Indicate the correct abbreviation for each enzyme.

Response5: The abbreviations of aspartate aminotransferase (AST/GOT), alanine aminotransferase (ALT/GPT) referred to:

Huang, X. J., Choi, Y. K., Im, H. S., Yarimaga, O., Yoon, E., & Kim, H. S. (2006). Aspartate aminotransferase (AST/GOT) and alanine aminotransferase (ALT/GPT) detection techniques. Sensors, 6(7), 756-782.

The abbreviations of lysozyme (LZM) referred to:

Klockars, M. A. T. T. I., & Reitamo, S. A. K. A. R. I. (1975). Tissue distribution of lysozyme in man.

The abbreviations of Na+K+ adenosinetriphosphatase (Na+/K+–ATPase), amylase(AMS), lipase(LPS), and trypsin(TPS) referred to:

Rajput, I. R., Li, Y. L., Xu, X., Huang, Y., Zhi, W. C., Yu, D. Y., & Li, W. (2013). Supplementary effects of Saccharomyces boulardii and Bacillus subtilis B10 on digestive enzyme activities, antioxidation capacity and blood homeostasis in broiler. International Journal of Agriculture and Biology, 15(2).

Point6: Most of the discussion is based on differences between after challenge and after recovery levels, however, were all those comparisons statistically significant? Statistics in the figures indicate differences between treatments for each sampling point, but what about differences for the same treatment between sampling points? For example, authors discuss lipase and amylase activity levels before and after the recovery, but activities patterns are so similar. From the results in the control group, it seems that those enzymes are not affected by stress, but are by dietary treatments. Therefore, for all the study, to obtain clear conclusions, comparisons between pre- and post- recovery are needed.

Response6: Thank you for your advice. Statistic about the differences for the same treatment between sampling points has been done, and all the figures have been redrawn to show the results.

Point7: Actualize references with recent bibliography on the topic:

Xavier, M.J. et al., 2022. Effects of dietary curcumin in growth performance, oxidative status and gut morphometry and function of gilthead seabream postlarvae. Aquaculture Reports 24, 101128. DOI: 10.1016/j.aqrep.2022.101128

Xavier, M.J. et al., 2021. Dietary curcumin promotes gilthead seabream larvae digestive capacity and modulates oxidative status. Animals 11(6), 1667. DOI: 10.3390/ani11061667

Response7: I have used references above into my discussion.[line 483-485]

Round 2

Reviewer 2 Report

Thank you for the response by the authors.

Well, the manuscript has been improved with respect to the presentation of data, but still, you need to make amendments in the English language which is not impressive. The very first sentence of the introduction is not clear to the readers. I suggest you to take the help of professional English editing services or alternatively consult a native English speaker.

Mu concerns are appended below:-

1. Thorghout the manuscript, mention "weight gain perecntage" instead of "weight gain rate %"

2. line 214 "no significant difference  is the same below"  this line is not clear. Rrepharse the sentences at other places in the manuscript.

3.  Rewrite the line 220.

4. I am not convinced by your reply on point 4. Could you please send me the excel data of the calculation made as  I believe the control value should have a constant value of 1 after the challenge and post recovery.   

Author Response

Response to Reviewer 2 Comments

(Round 2)

Point 1: Thorghout the manuscript, mention "weight gain perecntage" instead of "weight gain rate %"

Response 1: Thank you so much for your advice. All "weight gain rate %" have been replaced by "weight gain percentage".

Point 2: line 214 "no significant difference is the same below" this line is not clear. Rrepharse the sentences at other places in the manuscript.

Response 2: The mistake here has been revised in line 194-195.

Point 3: 3.  Rewrite the line 220.

Response 3: The description here has been rewritten into ‘The activity of Na+/K+–ATPase, γ-GT, and CK in intestine tissue was also determined.’[Line 219]

 Point 4: I am not convinced by your reply on point 4. Could you please send me the excel data of the calculation made as I believe the control value should have a constant value of 1 after the challenge and post recovery.   

Response 4: I’m so sorry for your confusion caused by us. The raw data will send to you.
